# In Silico Three-Dimensional (3D) Modeling of the SecY Protein of '*Candidatus* Phytoplasma Solani' Strains Associated with Grapevine "Bois Noir" and Its Possible Relationship with Strain Virulence

Roberto Pierro [1,2], Mariarosaria De Pascali [3], Alessandra Panattoni [1], Alessandro Passera [4], Alberto Materazzi [1], Luigi De Bellis [3], Andrea Luvisi [3,*], Piero Attilio Bianco [2,4] and Fabio Quaglino [4]

[1] Department of Agriculture, Food and Environment (DAFE), University of Pisa, Via del Borghetto 80, 56124 Pisa, Italy; rob.pierro@outlook.it (R.P.); alessandra.panattoni@unipi.it (A.P.); alberto.materazzi@unipi.it (A.M.)
[2] Institute for Sustainable Plant Protection, National Research Council (IPSP-CNR), Strada delle Cacce 73, 10135 Turin, Italy
[3] Department of Biological and Environmental Sciences and Technologies, University of Salento, Via Provinciale Monteroni, 73100 Lecce, Italy; mariarosaria.depascali@unisalento.it (M.D.P.); luigi.debellis@unisalento.it (L.D.B.)
[4] Department of Agricultural and Environmental Sciences, Production, Landscape, Agroenergy (DiSAA), University of Milan, Via Celoria 2, 20133 Milano, Italy; alessandro.passera@unimi.it (A.P.); piero.bianco@unimi.it (P.A.B.); fabio.quaglino@unimi.it (F.Q.)
* Correspondence: andrea.luvisi@unisalento.it

**Abstract:** Grapevine "bois noir", related to the presence of '*Candidatus* Phytoplasma solani' ('*Ca*. P. solani'), represents a serious threat in several vine-growing areas worldwide. In surveys conducted over two years, mild and/or moderate symptoms and lower pathogen titer were mainly associated with '*Ca*. P. solani' strains harboring a *secY* gene sequence variant (*secY*52), whereas severe symptoms and higher titer were mainly observed in grapevines infected by phytoplasma strains carrying any one of another four variants. A comparison of amino acid sequences of the protein SecY of '*Ca*. P. solani' strains revealed the presence of conservative and semi-conservative substitutions. The deduced three-dimensional (3D) protein analysis unveiled that one semi-conservative substitution identified in the sequence variant *secY*52 is responsible for a structural disordered region that probably confers a flexibility for binding to distinct molecular complexes. In fact, the other analyzed variants show an organized structure and the 3D in silico prediction allowed the identification of β-sheets. Thus, differences in symptom severity and pathogen concentration observed in grapevines infected by '*Ca*. P. solani' strains carrying distinct *secY* gene sequence variants suggest a possible relationship between SecY protein structure and phytoplasma strain virulence.

**Keywords:** bois noir; symptoms; *Vitis vinifera* cv. Sangiovese; grapevine yellows; protein prediction

## 1. Introduction

"Bois noir" (BN), related to '*Candidatus* Phytoplasma solani' (ribosomal subgroup 16SrXII-A) [1], is a disease belonging to the grapevine yellows (GY) complex. BN is largely present in Europe, the Mediterranean basin, Chile, South Africa, several countries of the Middle East, and China [2,3], constituting a major threat to viticulture in all main vine-growing areas. BN induces indistinguishable symptoms from other GY diseases, including berry shrivel, desiccation of inflorescences, uneven cane lignification, reddening or yellowing of leaves, reduction of growth, and general plant decline [4,5]. The polyphagous leafhopper *Hyalesthes obsoletus* (Hemiptera: Cixiidae) is the main insect vector involved in the phytoplasma transmission and erratically transmits it to grapevine [6]. Further studies

have also identified alternative plant hosts and insect species involved in the transmission of '*Ca.* P. solani' to grapevine [7–9], unveiling the possible existence of different BN epidemiological patterns.

The complexity related to the several host plants and insect vectors involved in the epidemiological patterns of phytoplasma strains made difficult the development of effective control strategies. Applications of antibiotics is not practicable in several countries, and it cannot represent an option for long time control [2]. The extensive use of pesticides against insect vectors and the removal of inoculum sources have been adopted in the management of phytoplasma diseases. Considering the inadequate results obtained in facing phytoplasma outbreaks worldwide and the negative impact on the environment and its biodiversity, these control strategies proved ineffective [5].

A novel sustainable approach to reduce the incidence of phytoplasma diseases can be developed by the investigation of genetic and biological aspects of phytoplasma strains and their interactions with plant hosts and vectors.

Genetic diversity among '*Ca.* P. solani' strains has been defined by sequence analyses of variable genes, such as *stamp*, *vmp1*, *secY*, and *tufB* [10–12]. Currently, '*Ca.* P. solani' molecular typing allowed identifying 59, 80, and 33 *stamp*, *vmp1*, and *secY* nucleotide sequence variants, respectively [13,14], while the sequence analyses of the gene *tufB* reported the presence of three '*Ca.* P. solani' *tuf*-types (a, b1 and b2), associated with different ecological systems [8–16]. In particular, the *secY* gene encodes an integral membrane protein constituting, along with SecE and SecG, a hetero-trimeric channel-like structure which participates in the SecY secretion system of preproteins through the bacterial cell membrane mediated by ATPase [17,18]. The SecYEG translocon is universally conserved and plays at the focus of bacterial protein transport, so as to regulate the translocation of proteins into and across the cytoplasmic membrane thanks to its ability to interact with multiple targeting factors, chaperones, and accessory proteins [19]. Prokaryotic Sec systems share some similarity elements, and non-synonymous amino acid substitutions of SecY secretion systems may lead to the emergence of new bacterial environment interactions, responsible for their adaptations to the host [20].

The present study aimed to analyze the *secY* gene nucleotide sequence variants among '*Ca.* P. solani' strains associated with different symptom severity in a cv. Sangiovese vineyard in Tuscany (Central Italy). The prediction and comparison of the three-dimensional (3D) protein structures were investigated and their possible relationship with the virulence of phytoplasma strains was investigated by bioinformatics tools.

## 2. Materials and Methods

### 2.1. Field Survey

In September 2018 and 2019, field surveys were conducted in a cv. Sangiovese vineyard (731 *Vitis vinifera*, L. 'Sangiovese' I-SS F9 A5 48) located in Greve in Chianti (Florence province, Tuscany, Central Italy). Each symptomatic grapevine was visually assessed and sorted according to GY symptom severity classes from 0 to 3, where class 0 was attributed to plants with no symptoms, class 1 to plants showing mild leaf symptoms on one shoot, class 2 to plants with leaf symptoms on two or three shoots, and class 3 to plants with leaf symptoms on more than three shoots and berry shrivel (Figure 1). About 3–10 symptomatic leaves were collected from each grapevine showing GY symptoms and stored at −20 °C until extraction of total nucleic acids.

Leaves collected from cv. Sangiovese plants located in the screen-house of the Department of Agriculture, Food and Environment (DAFE, University of Pisa, Italy) were utilized as healthy control, while leaves of periwinkle plants infected by phytoplasma strains STOL ('*Ca.* P. solani', subgroup 16SrXII-A, Acc. No. AF248959 [1], FD92 [Flavescence dorée phytoplasma, subgroup 16SrV-D, Acc. No. AF458380 [21], and AY (Aster yellows phytoplasma, subgroup 16SrI-B, Acc. N°. M30790) [22]), maintained in the greenhouse of the Department of Agricultural and Environmental Sciences (DiSAA, University of Milan, Italy) were used as infected controls.

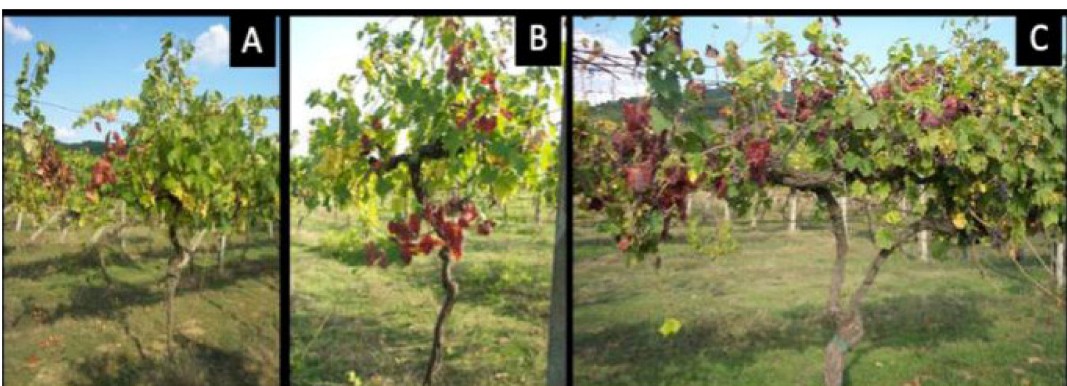

**Figure 1.** Grapevine classified in the symptom severity class 1 (**A**); class 2 (**B**) and class 3 (**C**).

*2.2. Total Nucleic Acids Extraction and Phytoplasma Detection*

For each plant, the midribs obtained from all the collected leaves were mixed, homogenized, and weighted to obtain a sample of 1 g. Total nucleic acids were extracted with 2% cetyltrimethylammonium bromide (CTAB)-based buffer from leaf veins [22], with some modifications, according to Pierro and colleagues [23]. Detection of phytoplasmas related to BN, FD, and AY was carried out through the specific TaqMan assay on the 16S ribosomal DNA using primer pairs and reaction conditions as described by Angelini et al. [24] in a Rotor-Gene Q thermal cycler (Qiagen, Germany). To determine if different genotypes were correlated to different phytoplasma abundance and/or symptom severity, relative quantification of phytoplasmas in each sample was calculated using the following formula: $\Delta Ct = Ct_p - Ct_g$, where $\Delta Ct$ is the normalized value, $Ct_p$ is the Ct obtained from amplification of phytoplasmatic 16S rRNA gene, and $Ct_g$ is the Ct obtained from amplification of the chaperonin gene.

*2.3. SecY Gene Amplification and Nucleotide Sequence Analysis*

The complete nucleotide sequence of *secY* gene was amplified using the primer pairs SecYF1a/SecYR1(XII), followed by SecYF2a/SecYR1(XII) [25], partially modified substituting degenerated positions with '*Ca*. P. solani' specific bases. In details, the degenerate positions M and Y of the primer SecYF2a (5′-CTCTTCGMCCYGGTTTTGAAGG-3′) were substituted with C and T, respectively. Mixtures and PCR conditions were carried out according to Lee and colleagues [22] in an automated thermal cycler C1000 Cycler Touch (Bio-Rad, USA). Healthy and infected controls were included in PCR assays. PCR products were analyzed through electrophoresis on 1% agarose gels in Tris borate-EDTA (TBE) buffer. *SecY* amplicons were sequenced in both strands (Sanger method) by a commercial service (Eurofins Genomics, Germany) and assembled by Contig Assembling Program of the software BioEdit 7.2.6 [26]. The nucleotide sequences identified in this study were trimmed to the annealing sites of the primers SecYF2a and SecYR1(XII), aligned by the ClustalW Multiple Alignment application, and searched for sequence identity by the Sequence Identity Matrix application of the software BioEdit. Based on sequence identity, *secY* gene nucleotide sequences of '*Ca*. P. solani' strains were grouped in distinct sequence variants. Using the dataset of *secY* gene variants based on the partial gene sequence (Supplementary Materials), *secY* sequence variants were trimmed to the same length and aligned with representative *secY* sequence variants to obtain a sequence identity matrix.

To determine the possible association among '*Ca*. P. solani' strains carrying distinct *secY* genetic variants, symptom severity classes observed in grapevines, and relative phytoplasma concentration, Chi square test and linear regression were carried out using SPSS statistical package for Windows, v. 24.0 (IBM Corporation, Armonk, NY) ($p < 0.01$).

*2.4. Protein Prediction and Model Evaluation*

Full sequences of *secY* variants identified in this study were analyzed in BLASTn and aligned with the complete '*Ca*. P. solani' *secY* sequences retrieved from NCBI Gen-Bank. SecY protein sequences were deduced from corresponding *secY* nucleotide sequences and analyzed using the Compute pI/MW tool (http://web.expasy.org/compute_pi/ accessed on 22 September 2021) for computation of theoretical isoelectric point and molecular protein weight. The amino acid sequences were aligned using the online CLUSTAL OMEGA program (https://www.ebi.ac.uk/Tools/msa/clustalo/ accessed on 22 September 2021) [27]. The SecY protein variants were also analyzed with the Motif Scan, TMPred, and SOSUI programs in order to analyze protein motifs and predict trans-membrane regions. The resulting 3D models were predicted using SWISS-MODEL programs [28] (http://swissmodel.expasy.org/ accessed on 29 September 2021) with the SecY protein of *Bacillus subtilis* as template (Swiss Model 6itc.1) and I-TASSER [29], a tool designed for multiple threading alignments and iterative structural assembly simulations. The structural models were rendered using UCSF-Chimera v 1.12 [30]. To evaluate the reliability of the modeled structures of SecY variants, different tools were used. The proteins structures were assessed for their reliability and model quality using the QMEAN server (http://swissmodel.expasy.org/docs/structure_assessment, accessed on 29 September 2021) [31]. Furthermore, 3D models were verified using PROCHECK [32] and ERRAT [33] programs. The prediction of active sites and possible ligand binding residues of SecY variants was generated using the COACH protein–ligand binding prediction server, a meta-server approach to protein-ligand binding site prediction (http://zhanglab.ccmb.med.umich.edu/COACH/ accessed on 4 October 2021). The complementary ligand binding site was predicted using COACH by matching the I-TASSER generated models of SecY protein variants with proteins in the BioLiP protein function database. Moreover, the functional templates were detected and ranked by COACH using composite scoring function based on structure and sequence profile comparisons [30]. The protein disorder analysis was carried out using DisEMBL (http://dis.embl.de/ accessed on 5 October 2021) [34] and PrDOS (http://prdos.hgc.jp/cgi-bin/top.cgi/ accessed on 5 October 2021) [35].

The MD simulations were carried out according to Pereira et al. (2019) [36] through the bioinformatics tool Galaxy [37,38].

## 3. Results

*3.1. Symptom Observation and Phytoplasma Relative Quantification*

In 2018, typical GY symptoms were observed in 66 out of 731 grapevines. Symptom severity classes 3, 2, and 1 were found in 42 (63.6%), 15 (22.7%), and nine (13.7%) out of 66 symptomatic grapevines, respectively (Figure 1). In 2019, GY symptoms were observed in 82 out of 731 grapevines. Symptom severity classes 3, 2, and 1 were found in 44 (53.6%), 28 (34.2%), and 10 (12.2%) out of 82 symptomatic grapevines, respectively (Figure 1).

Over the two consecutive years, all three symptom severity classes were observed in the examined vineyard and the disease incidence slightly increased from 9% (in 2018) to 11.2% (in 2019). Over the two years, the symptom severity class 3 was always the most represented, followed by the symptom severity classes 2 and 1 (Table 1). One grapevine assessed in the symptom severity class 1 in 2018 was symptomless in 2019, suggesting a symptom remission [5;15;23].

In accordance with a study reporting the occurrence of BN in the examined vineyard in previous years [23], a specific TaqMan assay carried out over the two consecutive years revealed the exclusive presence of '*Ca*. P. solani' in all symptomatic grapevines, excluding any infections caused by phytoplasmas associated with Flavescence dorée and Aster yellows. Real-time PCR analysis gave amplification from the periwinkle infected by the phytoplasma strain STOL, and there was no amplification from the periwinkles infected by the phytoplasma strains FD92 and AY, and the reaction mixtures devoid of nucleic acids or containing nucleic acids from healthy grapevine control plants. In accordance with previous results [23], relative quantification was carried out using ΔCt values determined

by the difference of Ct values obtained through the amplification of phytoplasma *16S* rRNA gene and the grapevine chaperonin gene (endogenous gene). In 2018 and 2019, lower ΔCts were associated with severe symptoms, while higher ΔCts were associated with mild or moderate symptoms. Based in this evidence, it can be assumed that a higher relative phytoplasma abundance was observed in plants infected by '*Ca.* P. solani' strains harboring the sequence variants *secY*1, *secY*6, *secY*9, and *secY*33, which were also associated with grapevines showing severe symptomatology. Conversely, a lower relative abundance of '*Ca.* P. solani' strains harboring the sequence variant *secY*52 was associated with grapevines showing mild symptoms (Figures 2 and 3).

**Table 1.** Prevalence of *secY* gene sequence variants, carried by '*Ca.* P. solani' strains identified in 2018 and 2019, in vines showing different symptom severity (average).

| No. of '*Ca.* P. Solani' Strains Collected | | | Symptom Severity | Relative Abundance | Symptom Severity | Relative Abundance |
|---|---|---|---|---|---|---|
| **2018** | **2019** | **Sequence Variant** | **(Average 2018)** | **(Average 2018)** | **(Average 2019)** | **(Average 2019)** |
| 16 | 19 | *secY*1 | 2.8 | 8.0 | 2.6 | 8.7 |
| 7 | 12 | *secY*6 | 2.7 | 6.4 | 2.6 | 5.8 |
| 20 | 26 | *secY*9 | 2.6 | 6.2 | 2.5 | 7.1 |
| 11 | 7 | *secY*33 | 2.6 | 8.9 | 2.8 | 8.3 |
| 12 | 18 | *secY*52 | 1.6** | 12.8 | 1.4 ** | 13.1 |

**: significantly different distribution (Chi square test, $p < 0.01$).

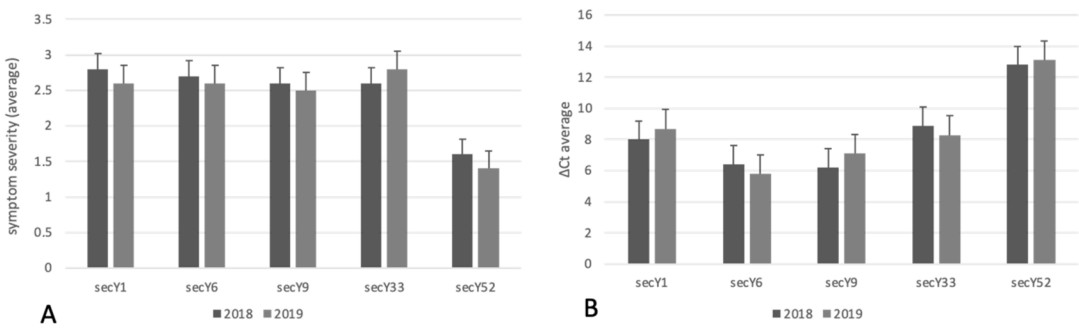

**Figure 2.** Count of '*Ca.* P. solani' strains with different *secY* sequence variants identified in grapevines showing different symptom severity (**A**) and relative concentrations (**B**).

Linear regression, carried out using ΔCt values obtained from '*Ca.* P. solani' strains harboring different sequence variants and symptom severity classes, showed statistically significance differences in the relative concentrations of phytoplasma in grapevines exhibiting mild, moderate, and severe symptomatology ($p < 0.01$) (Figure 3). The chi square test showed the statistically different distribution of Ca. P. solani strains associated with mild, moderate, and severe symptoms and *secY* sequence variants (Table 1).

### 3.2. Identification of secY Sequence Variants

Specific nested PCR assays allowed the amplification of complete *secY* gene from all infected grapevine samples.

The nucleotide sequence analysis of the s*ecY* gene detected the presence of five distinct sequence variants among '*Ca.* P. solani' strains analyzed over the two consecutive years (Table 1). Four out of these five variants, trimmed to equal length and aligned with partial *secY* gene sequence variants previously described (Supplementary Table S1), resulted undistinguishable from the variants *secY*1, *secY*6, *secY*9, and *secY*33, [14]. The fifth variant, here named *secY*52, represented a novel *secY* variant. Each *secY* sequence variant identified in this study was deposited to NCBI GenBank at Accession numbers: MW051357 (variant *secY*1), MW051358 (*secY*6), MW051359 (*secY*9), MW051360 (*secY*33), and MW051361 (*secY*52).

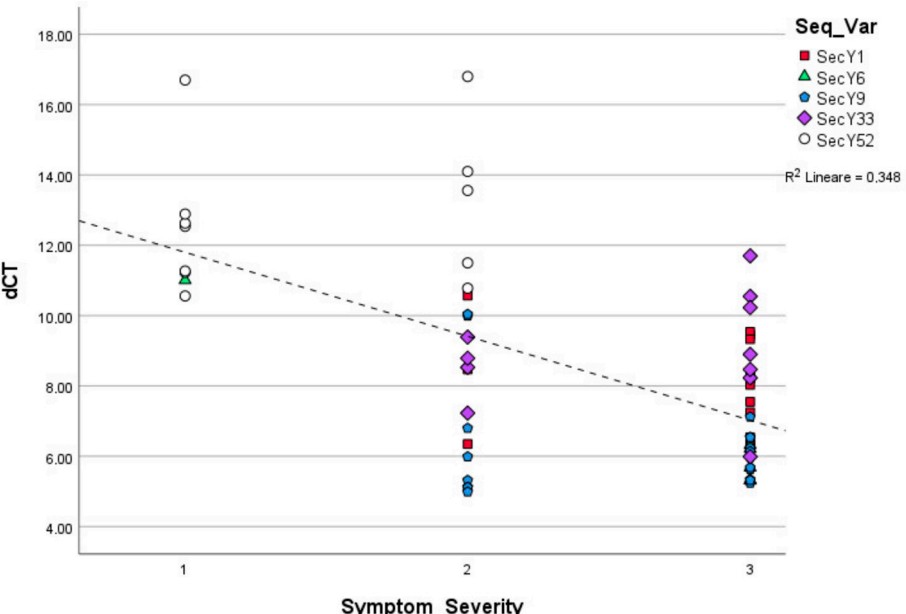

**Figure 3.** Linear regression among 'Ca. P. solani' strains carrying distinct secY sequence variants symptom severity classes observed in grapevines and relative phytoplasma concentration ($p = 0.001$).

BLASTn analysis, carried out using the full sequence of the gene *secY*, showed that the variants *secY6*, *secY9*, *secY33*, and *secY52* shared a sequence similarity that ranged from 99.7% to 100% with 'Ca. P. solani' strains EC-KH-Defzol (Acc. N°KX685881) and P10 (Acc. N°. KU374893) previously found in previously found in Iran in *Eucalyptus* sp. and Bosnia and Herzegovina in *Capsicum annuum*, respectively, while the variant *secY1* shared 100% sequence homology with 'Ca. P. solani' strain M7 (Acc. N°. KU374896), already detected in Bosnia and Herzegovina in *Zea mays*.

In 2018, the prevalent *secY* sequence variant was *secY9* (identified in 20 out of 66 'Ca. P. solani'-infected grapevines), followed by *secY1* (16 grapevines), *secY52* (12 grapevines), *secY33* (11 grapevines), and *secY6* (7 grapevines). In 2019, the prevalent *secY* sequence variant was *secY9* (26 grapevines), followed by *secY1* (19 grapevines), *secY52* (18 grapevines), *secY6* (12 grapevines), and *secY33* (7 grapevines) (Table 1).

Interestingly, the chi square test showed statistically significance differences in the distribution of 'Ca. P. solani' strains carrying distinct *secY* sequence variants in grapevines showing different symptom severity ($p = 0.004$), suggesting that phytoplasmas harboring *secY52* sequence variant were associated with mild and/or moderate symptoms (1.6 and 1.4 average symptom severity in 2018 and 2019, respectively) (Table 1). Differently, severe symptoms were mainly observed in vines infected by phytoplasma strains carrying the *secY1*, *secY6*, *secY9*, and *secY33* genetic variants (Table 1; Figure 2).

### 3.3. Protein Structure Predictions

Analyses of in silico translated SecY amino acid sequences allowed the identification of five SecY protein variants (407 aa), corresponding to the *secY* gene nucleotide sequence variants. For all sequences, the application of BLASTp search comparison tool revealed the higher identity (>99%) with 'Ca. P. solani' SecY translocase preprotein (Acc. N°. WP023161280).

Since the amino acid sequences of SecY1 and SecY9 are identical, amino acid alignment carried out using CLUSTAL OMEGA program distinguished four SecY variants. The variant SecY6 differs from the previous two sequences for the presence of asparagine instead of aspartic acid in position 123 (D (aspartic acid) → N (asparagine)], while SecY33 diverges as leucine is replaced by phenylalanine in position 77 [L (leucine) → F (phenylalanine)). Lastly, SecY52 shows asparagine but not serine in position 292 (S (serine) → N (asparagine)) (Figure 4).

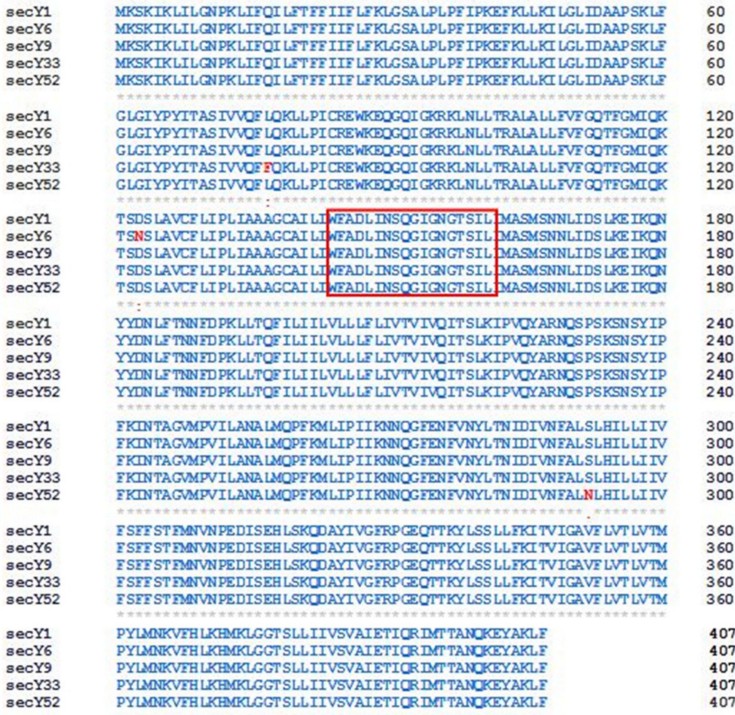

**Figure 4.** Sequence alignment of SecY protein variants identified in the present study. Red colored box indicates the SecY signature [39]. Identical residues '*'; conserved substitutions (similarity > 70%) ':'; semi-conserved substitutions (35–70% similarity) '.'. Numbers indicate amino acid positions.

Analyses carried out with TMPred and SOSUI indicate that the SecY protein variants have 10 transmembrane segments. In particular, the semi-conservative substitution S (serine) → N (asparagine) in position 292 of the SecY52 variant collects in the seventh transmembrane domain (transmembrane region 7), which is directly involved in the opening/closing regulation of the SecY translocation channel (Figure 5) [40,41].

The 3D models of SecY proteins were obtained by comparing the results provided by SWISS-MODEL [28] and I-TASSER [29]. These programs can model the protein structures respectively via comparative modeling and threading. To date, comparative modeling is the most successful and accurate method. However, searching for homologous proteins is difficult when the sequence identity falls between 10% and 30% [42]. In this case, the sequence identity of the SecY proteins variants with available structures in PDB is less than 30% (Supplementary Table S3), this increases the probability of errors in predicted models, such as errors in side-chain packing, distortions and shifts in correctly aligned regions, errors in regions without a template, errors due to misalignment and incorrect templates [43]. Furthermore, QMEAN Z-score was −4.48, −4.44, −4.42, −4.42, and −4.48 for SecY52, SecY33, SecY9, SecY1, and SecY6, respectively. The QMEAN Z-score provided an estimation of the "degree of nativeness" of the structural features observed in the model on a global scale [31]. A QMEAN Z-score value of approximately zero specifies superior quality between the modeled structure and experimental structures. The obtained scores indicate models characterized by lower quality. However, this result is due to the template structure from which they were built as there are no fully suitable templates available. This limit is closely related to difficult in vitro phytoplasma cultivation. The SecY proteins models also were verified using the Ramachandran plot from the MolProbity program (Supplementary Figure S1). Therefore, to validate the obtained 3D models of SecY proteins, a threading approach has been used in order to obtain the most accurate structure (Supplementary Table S4). Initially, ten models for each sequence were obtained using threading templates from the PDB structural database according to their Z score values ranging from 2.50 to 6.18. From them, the two best models were selected with C

score values ranging from 0.85 to 0.02 (SecY52), 0.98 to 0.01 (SecY6), 0.50 to −1.44 (SecY33), 0.92 to 0.01 (SecY9), and 0.95 to 0.01 (SecY1). The optimized models were evaluated by PROCHECK [32] and ERRAT [33] programs (Table 2), obtaining satisfactory evaluations.

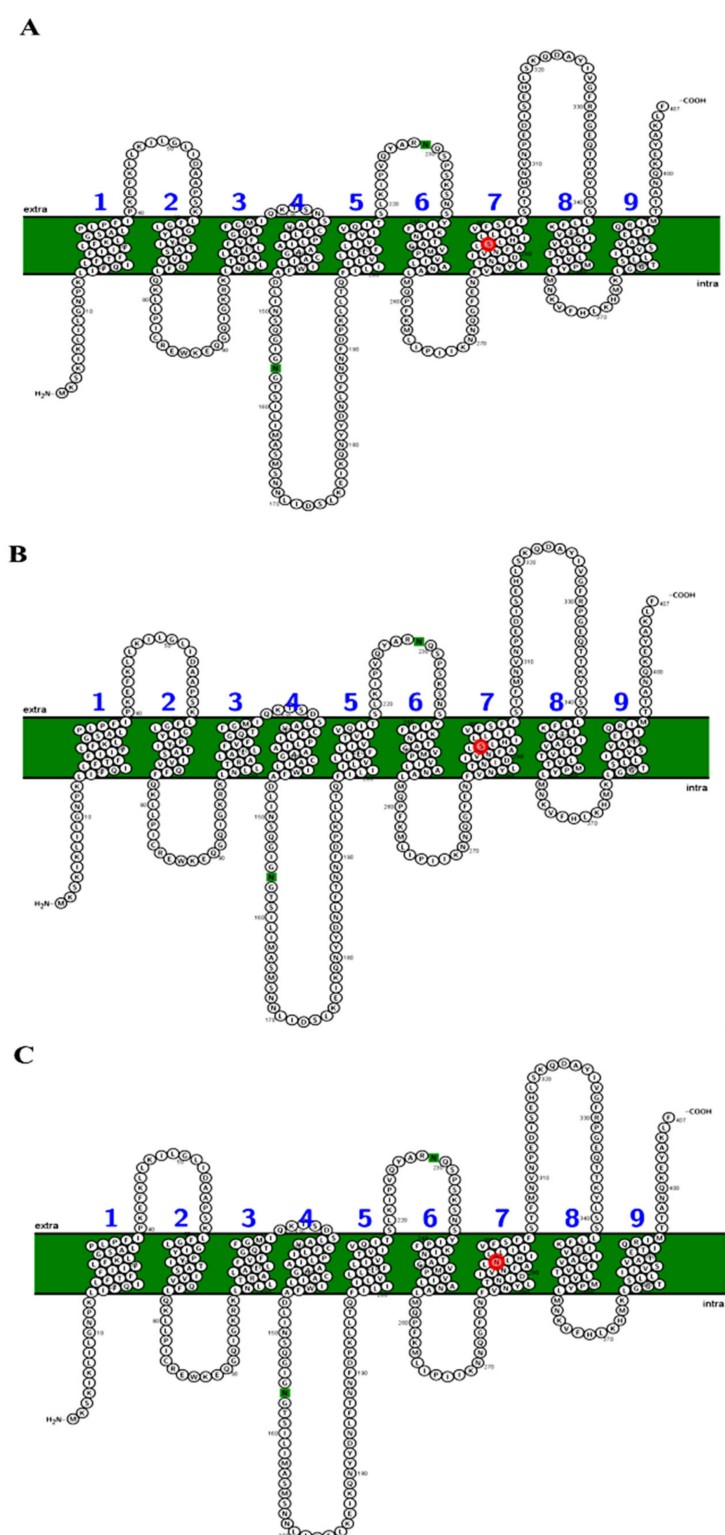

**Figure 5.** Schematic representation of the organization of the SecY6 (**A**), SecY33 (**B**), and SecY52 (**C**) variant structures consisting of 10 transmembrane domains. The single amino acid substitution in transmembrane region 7 region the position 292 is marked in red. Each amino acid is abbreviated by its first-letter code, green circle indicates signal peptide.

**Table 2.** Assessment of the predicted three-dimensional structures of SecY proteins by PROCHEK and ERRAT programs.

| | Validation Index | |
| --- | --- | --- |
| **SecY Sequence Variant** | **PROCHECK** | **ERRAT** |
| SecY1 | 88.8% | 86.16% |
| SecY6 | 88.5% | 85.37% |
| SecY9 | 88.8% | 86.16% |
| SecY33 | 89.1% | 82.59% |
| SecY52 | 69.8% | 59.46% |

A comparison among the predicted 3D structures of the SecY proteins showed that the replaced amino acids are all located at the surface of the folded proteins, suggesting possible functional implications (Figure 5). The program predicted all residues involved in the formation of active site pockets for ligands.

All ligand binding sites predicted by COACH server have C-scores close to 1, indicating more reliable predictions (Figure 6).

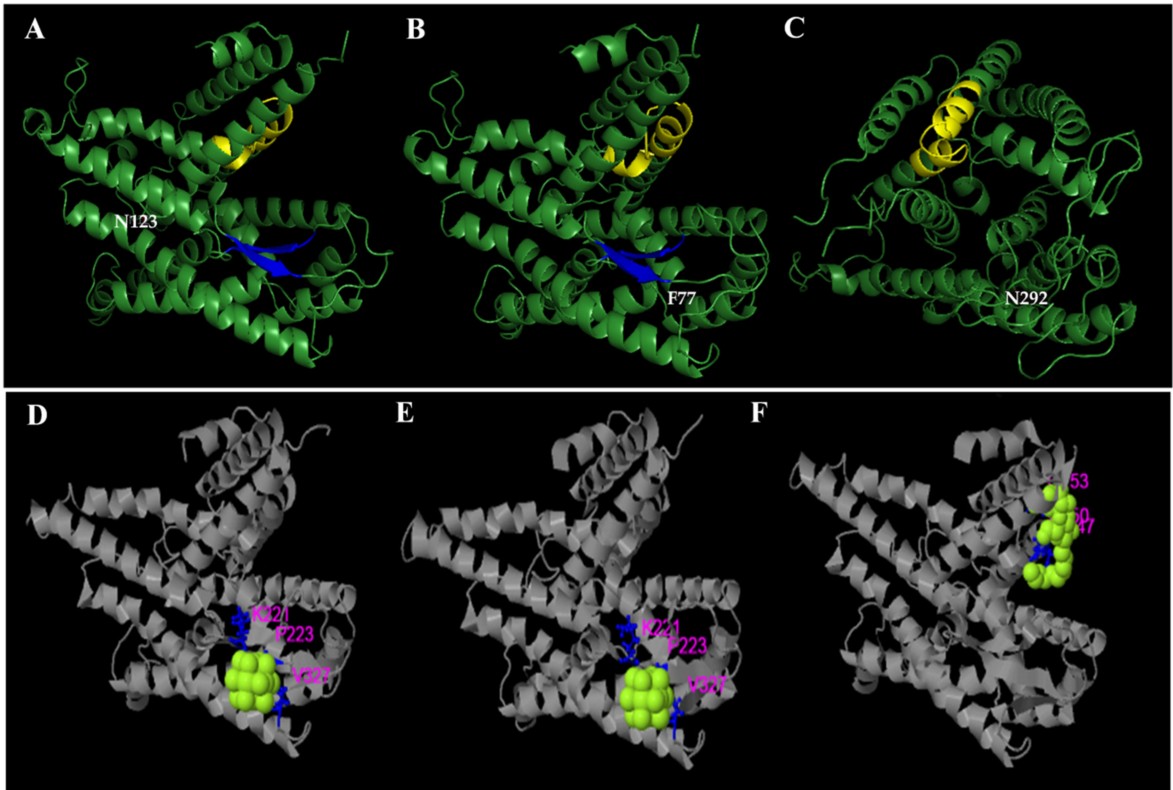

**Figure 6.** Structural comparison between SecY6 (**A**), SecY33 (**B**), and SecY52 (**C**) protein variants. Three-dimensional structures are modelled based on the sequence alignment with *Bacillus subtilis* SecY (6itc.1). Amino acid residues that differ in the strains are tagged in white. The yellow and blue colors indicate respectively the protein SecY signatures domain and the beta-sheet structure. The predicted binding ligand is highlighted in yellow-green spheres for SecY6 (**D**), SecY33 (**E**), and SecY52 (**F**), with the corresponding binding residues shown as blue ball-and-stick illustrations in the picture of the 3-D model.

Although the analysis reveals only one semi-conservative substitution in the SecY52 variant, it is interesting to observe that its 3D structure is different in comparison with the variants, namely SecY1, SecY6, SecY9, and SecY33, which have a more ordered structure

and exhibit a beta-sheet structure (Figure 7). Thus, the replacement of a single amino acid apparently induces a substantial modification in the 3D structure of the SecY52 variant.

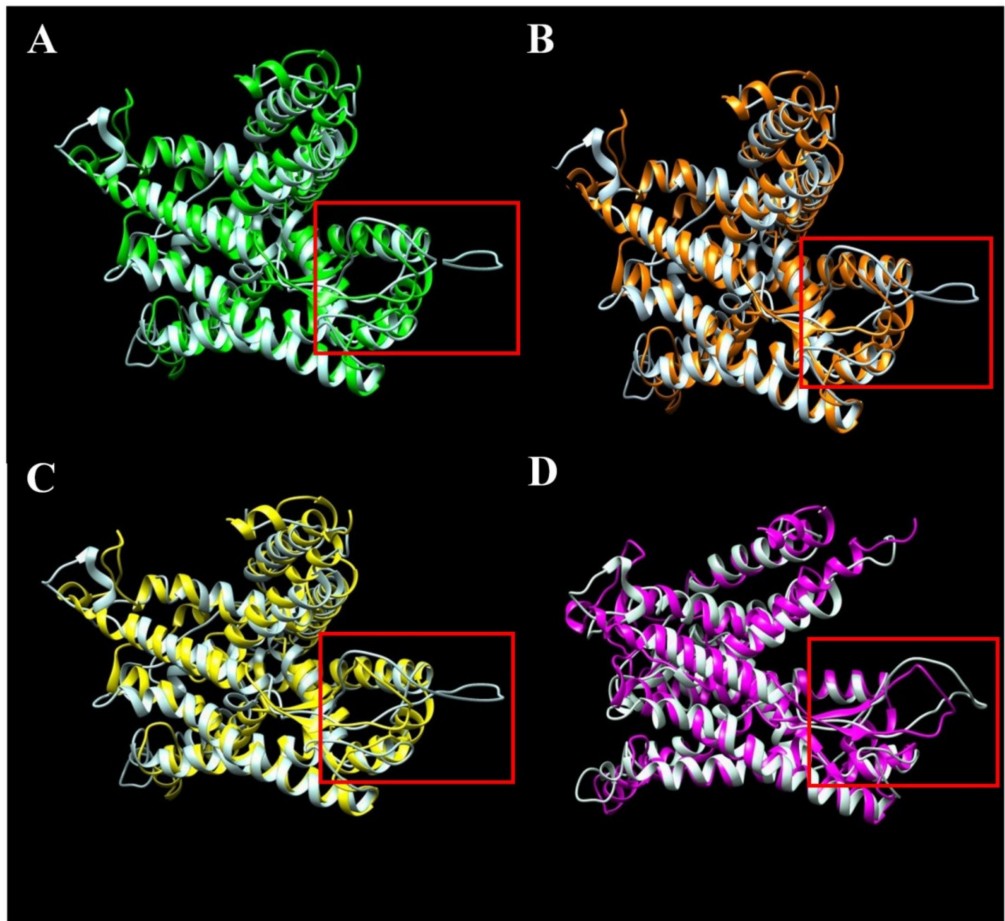

**Figure 7.** Structural comparison through overlap of semi-conservative variant SecY52 colored in white with conservative variants SecY1 (**A**), SecY6 (**B**), SecY9 (**C**), and SecY33 (**D**). Three-dimensional structures were modelled based on the sequence alignment with *Bacillus subtilis* SecY (6itc.1) (Supplementary Figure S2). The hypothetical disordered region is indicated in the red colored box.

Multiple programs are currently available for the prediction of intrinsic disorder regions in protein sequences, which, in general, are based on the on the principle that structurally disordered regions contain a principally of charged and hydrophilic residues, and showed lower sequence complexity.

In this work, to predict disordered regions of SecY variants, DisEMBL [34] and Pr-DOS [35] programs were used (Figure 8).

We conducted molecular dynamics (MD) simulations of the variants SecY to understand the role of these amino acid substitutions on protein structure. The MD simulations were performed in according to the methodology described by Pereira et al. [36,44,45]. An analysis of the GROMACS [46] simulation outputs (structure and trajectory file) was carried out using Galaxy tools (https://cheminformatics.usegalaxy.eu, accessed on 15 October 2021) developed for computational analysis (Supplementary Figures S3 and S4) [37,38].

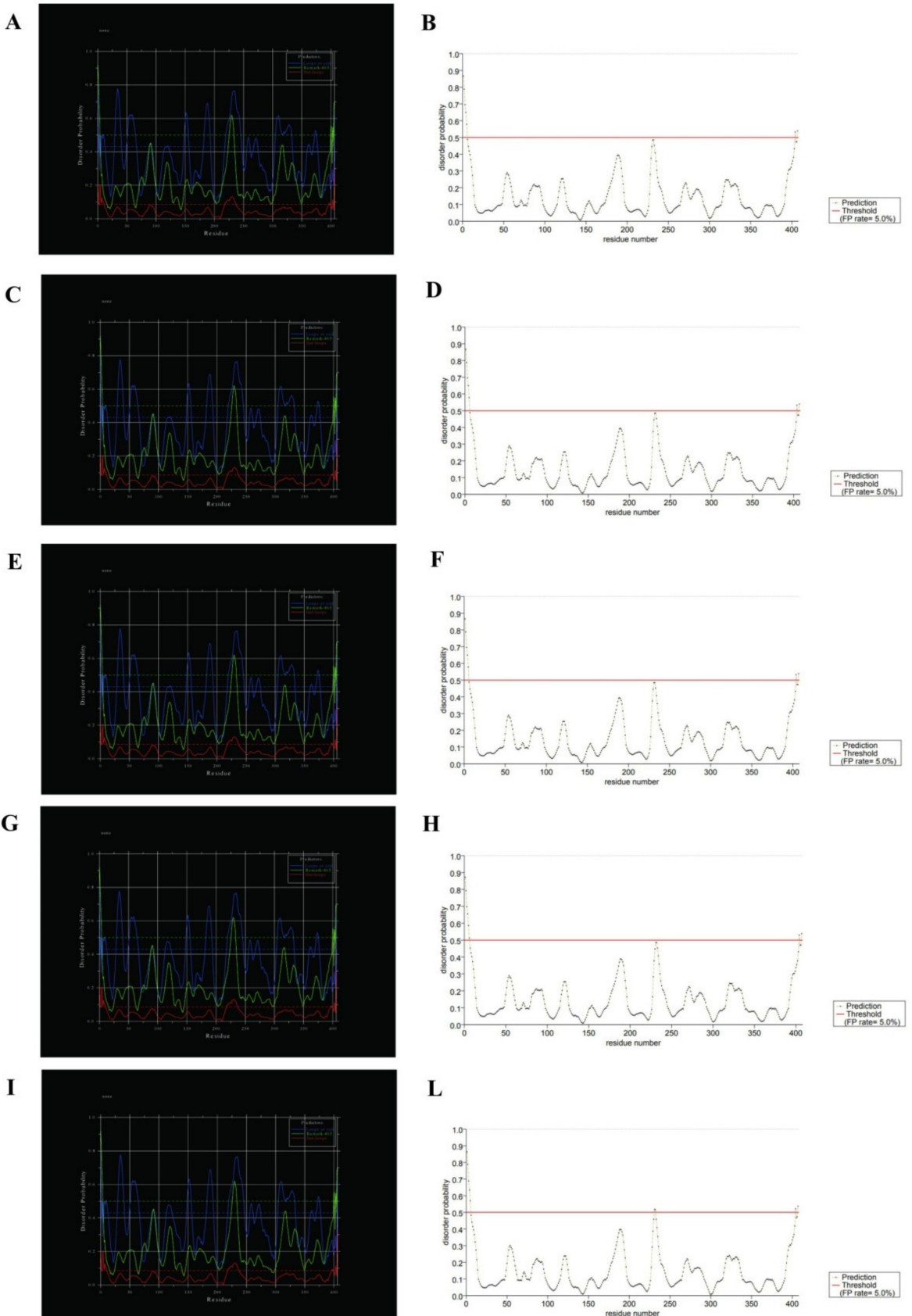

**Figure 8.** Disorder prediction for SecY variants by DisEMBL and PrDOS programs. To the left SecY1 (**A**), SecY6 (**C**), SecY9 (**E**), SecY33 (**G**), and SecY52 (**I**) was analyzed through DisEMBL program; to the right SecY1 (**B**), SecY6 (**D**), SecY9 (**F**), SecY33 (**H**), and SecY52 (**L**) was analyzed through PrDOS.

## 4. Discussion

The 3D in silico prediction allowed the identification of β-sheets in the folded structures of SecY1, SecY6, SecY9, and SecY33 protein variants, which play a critical role in determining protein stability and biomolecular recognition [47,48], whereas the variant SecY52 showed a disordered structure and has no β-sheet.

The β-sheet structure is conserved in the SecY protein and stabilizes the bond with the SecA subunit in order to form the Sec system [49]. It is known that the β-sheets structures are inherently aggregation prone, because they are in the right conformation to interact with any other β strand. In recent years, protein aggregation through β-sheet interactions has increasingly drawn attention because it occurs in many serious human diseases [50,51]. Our knowledge there are to no reports of a correlation between protein aggregation and disease in plants, although, it is known that the stress responses result in increased protein aggregate formation at the cellular level [52].

Furthermore, historically, the functionality of a protein was correlated by its unique 3D structure. Recently, several researchers have pointed out that the lack of structure or flexibility can be important for biological function [53–56]. The current view is that disordered proteins, thus characterized by a lack of stable three-dimensional structure, frequently interact with many partners or function as hubs in protein interaction networks [57–59]. Recent and extensive research has provided evidence that many proteins lack fixed structure (are disordered) under physiological conditions, and their functions depend on the unstructured rather than the structured state [60]. It has also been demonstrated that protein disorder plays a key-role in biology and in diseases mediated by protein misfolding and aggregation [61], as the function is directly linked to structural disorder [62,63]. Interestingly, the single amino acid substitution observed in the variant associated with milder symptoms (SecY52) concerns a Ser292 in transmembrane region 7, a functionally important region in secretory-protein export through the Sec pathway. Transmembrane regions mainly play two roles: some transmembrane regions may only anchor the protein to the membrane with correct local orientations, others may have direct functional roles. In particular, the transmembrane region 7 near the periplasm was associated with channel gating and lateral movement of a signal sequence. Furthermore, recently, similar substitutions have been shown to have a functional role in *Escherichia coli* [64]. Specifically, bacterial cells containing the mutation are resistant to toxins by blocking their movement into and through target cell membranes. It is known that SecY interacts with SecA and SecE in a protein translocation system (Sec-dependent pathway) involved in the protein sorting process [49]. It was reported that antigenic membrane proteins and phytoplasma effectors are sorted by the Sec-dependent pathway to the phytoplasma cell surface and host cell cytoplasm, playing a crucial role in plant–pathogen molecular interaction [65]. Considering that (i) statistically significant differences in the distribution of '*Ca*. P. solani' strains carrying distinct *secY* sequence variants in grapevines with different symptom severity were found (Table 1), (ii) statistically significant differences in the relative phytoplasma concentration carrying distinct *secY* sequence variants in grapevines with different symptom severity were found, and that (iii) SecY52, associated with mild and/or moderate symptoms, differs by a specific amino acid alteration located in transmembrane region 7 from the other SecY protein variants harbored by '*Ca*. P. solani' strains affecting grapevines showing severe symptoms, it is reasonable to hypothesize that structural modifications in the protein SecY52, possibly associated with the altered function of the Sec-dependent pathway, may be related to hypo-virulent '*Ca*. P. solani' strains. This hypothesis can be reinforced by previous studies highlighting the relationship between '*Ca*. P. mali' strain virulence and diversity in the structure of AAA plus ATPases and HflB/FtsH proteases [66].

## 5. Conclusions

Grapevine-infecting '*Ca*. P. solani' strains harboring the *secY*52 sequence variant have a single amino acid substitution in the functional transmembrane region 7 and the absence of a β-sheet structure within the SecY protein translocase. For two consecutive

years, such strains were correlated to mild/moderate grapevine yellows symptoms and low phytoplasma concentration in infected plants. These results, along with previous evidence reporting a possible range of virulence among '*Ca*. P. solani' strains associated with distinct *stamp* gene sequence variants [13,14], suggest that multiple genes could determine '*Ca*. P. solani' intra-specific differences in the interaction with the plant hosts. Starting from such evidence, it would be interesting to expand the analysis of *secY* and other genes (i.e., transmembrane proteins and virulence factors) to '*Ca*. P. solani' strains infecting a range of grapevine cultivars with different symptom severity from diverse geographic areas. Using this approach, it could also be interesting to extend such analysis on phytoplasma strains identified in weeds and insect vectors to investigate possible specific epidemiological patterns related to phytoplasma causing a different range of symptom severity in grapevines.

**Supplementary Materials:** The following supporting information can be downloaded at: https://www.mdpi.com/article/10.3390/ijpb13020004/s1, Figure S1: Ramachandran plot of modelled protein of secY1 (A); secY6 (B); secY9 (C); secY33(D); secY52 (E). Figure S2. Alignment of the SecY protein variants and the template sequence 6itc.1 of *Bacillus subtilis*. Figure S3 Molecular dynamics (MD) simulations of SecY variant structures. The root-mean-square deviation (RMSD) timeseries for the variants SecY: SecY1 (A), SecY6 (B), SecY9 (C), SecY33(D) and SecY52 (E). Figure S4 Molecular dynamics (MD) simulations of SecY variant structures. The root-mean-square fluctuation (RMSF) vs. the residue position for the variants SecY: SecY1 (A), SecY6 (B), SecY9 (C), SecY33(D) and SecY52 (E). Table S1: Sequence variants Dataset of the gene *secY* among '*Ca*. P. solani' strains available in GenBank. Table S2: Sequence variants Dataset of the preotein SecY among '*Ca*. P. solani' strains available in GenBank. Table S3. Sequence identity of SecY variants with templates in Protein Data Bank (PDB). Table S4. Top 10 threading templates used by I-TASSER.

**Author Contributions:** Conceptualization, L.D.B., A.M., A.L., P.A.B. and F.Q.; data curation, R.P., M.D.P., A.P. (Alessandro Passera) and A.L.; formal analysis, R.P. and M.D.P.; investigation, R.P., A.P. (Alessandra Panattoni), A.P. (Alessandro Passera) and A.M.; methodology, R.P., M.D.P. and A.P. (Alessandra Panattoni); software, R.P. and M.D.P.; writing—original draft, R.P. and M.D.P.; writing—review & editing, A.L., L.D.B. and F.Q. All authors have read and agreed to the published version of the manuscript.

**Funding:** This research received no external funding.

**Institutional Review Board Statement:** Not applicable.

**Informed Consent Statement:** Not applicable.

**Conflicts of Interest:** The authors declare no conflict of interest.

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
