# Peer review of "In Silico Three-Dimensional (3D) Modeling of the SecY Protein of ‘Candidatus Phytoplasma Solani’ Strains Associated with Grapevine “Bois Noir” and Its Possible Relationship with Strain Virulence"

_2037-0164, doi:10.3390/ijpb13020004_

Round 1

Reviewer 1 Report

The findings described are extremely interesting, but overall I think that more information on the materials and methods used are needed to support them.

Major comments: My major comments are related to the relative quantification of phytoplasma in the plant, and the assumption by the authors that "higher relative phytoplasma abundance was associated with grapevines showing severe symptomatology..." (lines 184-185). While the results show a lower relative concentration of SecY52 isolates compared to other SecY variants, the overall assumption that higher relative phytoplasma abundance was associated with grapevines showing severe symptomatology is not evident to me.

  • (line 83) The authors should clarify whether all the plants in the vineyard are the same Sangiovese clone, or different clones.
  • (lines 89-90) The authors should clarify if the 10-12 leaves collected per plant were all symptomatic, or collected from symptomatic shoot(s), or collected randomly from each symptomatic plant (therefore including both symptomatic and asymptomatic leaves). In view of the erratic distribution of phytoplasmas in the host palnts, this should be considered when comparing the relative phytoplasma concentration and it is especially critical in the case of plants assessed as "class 1" (mild leaf symptoms on one shoot), since only a limited number of leaves would show symptoms as depicted in Fig1a.
  • (lines 101-103) The cited manuscript [23] describe using 1 g of midribs dor DNA extraction. Were all the 10-12 leaves collected from each plant used to obtain 1g of midribs? Were the midribs obtained from 10-12 leaves homogenised before subsampling 1 g for DNA extraction? This is important especially if both symptomatic and asymptomatic leaves were collected from the same plant (see previous comment).
  • (lines 182-187) Tab1 and Fig2 indicate an association between symptoms severity and SecY genotype, not symptoms severity. Please double-check and correct accordingly the text and/or the table and figure.
  • (lines 166-171 and Table1, discussion section) The authors should better describe and explain the variation in the number positive plants across the two years and how it can/cannot be explained by remission of symptoms. More in general, remission of symptoms (which often occurs) is not considered when discussing the findings.
  • Section 3.1 "Symptom observation and phytoplasma detection". In this section the authors also describe the relative quantification of Ca. P. solani. Please add "relative quantification" to the title of this section, or consider creating an additional section to describe the results of relative quantification.
  • (lines 188-191 and Fig3) Looking at Fig3, I would be cautions to correlate the deltaCt value with the symptoms severity. Rather, the SecY52 genotype seems to be associated with higher deltaCt values (which is already shown in Tab1 and Fig2). If you remove the SecY52 datapoints from  the plot, I wouldn't think there is any correlation between symptoms severity and deltaCt value.

Minor comments:

  • (line 50) "The huge complexity...", please consider changing"huge" for a more scientifically-sound term
  • (line 120) "SecY amplicons were sequenced in both strands (5× coverage, Sanger method)..." If the amplicons were sequenced in both strands, how can the coverage be 5x? please doublecheck. Also, it is not clear to me why it is necessary to specify the coverage for sequences obtained by sanger sequencing of PCR amplicons.

Reviewer 2 Report

This is a carefully done study and the findings are of considerable interest. Purpose of the present manuscript is "In silico three-dimensional (3D) modeling of the SecY protein of ‘Candidatus Phytoplasma solani’ strains associated with grapevine “bois noir” and its possible relationship with strain virulence".

This article is interesting, useful, and well prepared. In conclusion grapevine-infecting ‘Ca. P. solani’ strains harboring the secY52 sequence variant have a single amino acid substitution in the functional transmembrane region 7 and the absence of a β-sheet structure within the SecY protein translocase. For two consecutive years such strains were correlated to mild/moderate grapevine yellows symptoms and low phytoplasma concentration in infected plants.

Introduction it is in line with the Instructions for the Authors. The methodology is also corresponding to the experimental part points of interest. Results are representative and comprehensive.

Discussion is also appropriate, but a more precise direction of future research should add more value.

References are in line with the scientific demonstration of the main issues.

This is a carefully done study and the findings are of considerable interest. 

Round 2

Reviewer 1 Report

I thank the Authors for their reply letter and for having made corrections that addressed my previous comments.

Overall, I think the manuscript has been considerably improved.

I have on more consideration that the authors might need to address:

  • point 6 of the Authors' Reply Letter: The authors expanded the text of the manuscript in section 3.1, stating that "One grapevine assessed in the symptom severity class 1 in 2018 was symptomless 2019, suggesting a symptom remission". However, the authors are still not considering the change in numbers of plants infected by different SecY variants between the two years.  As an example, there were 11 plants infected with SecY33 in 2018, and only 7 infected with the same SecY variant in 2019 (table 1). Does this mean that 4 of them did not show symptoms and were not tested in 2019? Or they were found infected by a different SecY variant in 2019? (of course the same applies for the other SecY variants as well)
